# Synthesis of Annotated Colorectal Cancer Tissue Images from Gland Layout

**Editor:** Unnamed

## Abstract

Generating annotated pairs of realistic tissue images along with their annotations is a challenging task in computational histopathology. Such synthetic images and their annotations can be useful in training and evaluation of algorithms in the domain of computational pathology. To address this, we present an interactive framework to generate pairs of realistic colorectal cancer histology images with corresponding tissue component masks from the input gland layout. The framework shows the ability to generate realistic qualitative tissue images preserving morphological characteristics including stroma, goblet cells and glandular lumen. We show the appearance of glands can be controlled by user inputs such as number of glands, their locations and sizes. We also validate the quality of generated annotated pair with help of the gland segmentation algorithm.

**Keywords:** Computational Pathology, Generative Adversarial Networks, Tissue Image Synthesis

## 1. Introduction

Deep learning techniques have shown great potential in solving problems in the field of computational histopathology such as tumor segmentation (Bejnordi et al. (2017)), nuclei classification (Tripathi and Singh (2020)) and cancer grading (Gupta et al. (2019); Shaban et al. (2020)), particularly for facilitating clinical diagnosis. These applications generally required a large amount of training data which is difficult to acquire, requires involvement of highly trained pathologists and can be a time-consuming and costly process. Furthermore, there are several confidentiality issues relating to patients, legal & ethical barriers while collecting pathology images (Price and Cohen (2019), Chen et al. (2021)). Such developments compelled researchers to come up with novel solutions for generation of high-quality pathology images (Kovacheva et al. (2016), Quiros et al. (2019), Levine et al. (2020)).

In pathology image analysis, due to tissue heterogeneity and variations in acquired tissue images in laboratories, the data annotation phase must often be repeated for different tissue types such as glands, fat tissues, blood vessels to achieve the optimal performance. For tasks like gland segmentation, manual generation of component masks highlighting glandular portions, can be a laborious and time-consuming task. This is a major obstacle to generating a large amount of annotated data for segmentation algorithms. Some researchers have investigated generating synthetic pathology images from tissue component masks. For instance, Senaras et al. (2018) proposed a model to generate breast cancer tissues conditioned on the input nuclei masks; Deshpande et al. (2020) presented the framework to produce high resolution colorectal tissue images based on glandular masks. Hou et al. (2019) proposed an unsupervised pipeline to construct histopathology tissue images from corresponding nuclei

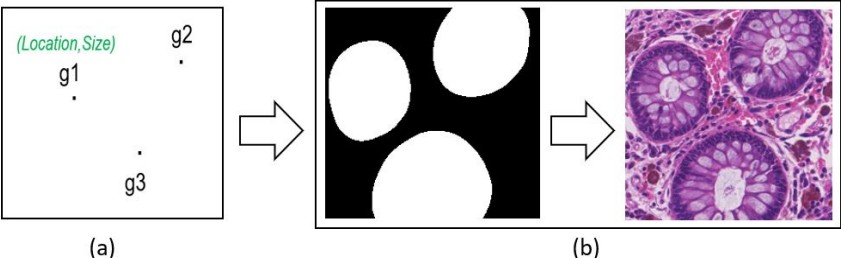

Figure 1: Concept diagram of the proposed framework. The framework generates annotated pairs (b) of colorectal tissue images with their corresponding tissue component (gland) masks from the input gland layout (a). The gland layout is a 2-d plane where users can arrange glands on 2-d locations with suitable sizes. The generated annotated pairs can be used in training and evaluation of the gland segmentation algorithm.

masks constructed using the predefined random polygon generator. The pairs were used to train the nuclei segmentation algorithm. These methods either assume input component masks are already present, or require explicit construction of component masks by generating random shapes of respective tissue components like nuclei, which can be erroneous and may not be realistic. Moreover, this process of crafting component masks can be tricky for larger multi-cellular structures such as glands. Generating synthetic images along with component masks simultaneously is therefore desirable as it potentially reduces the cost of annotations and also constructs realistic annotated pairs.

In this work, we propose a user-interactive framework that can generate colorectal tissue images along with corresponding tissue component masks simultaneously from the input gland layout, a layout where user can specify locations and sizes of the glands in colorectal tissue images. The conceptual view of our idea can be seen in Figure 1. We leverage properties of GANs under an adversarial setting to produce realistic tissue images. Key highlights of our work are:

1. We propose a framework that can generate realistic colorectal tissue images and corresponding tissue component/gland masks simultaneously.

2. The proposed framework allows user to change the appearance of glands by their locations and sizes. To the best of our knowledge, it is the first framework that can generate annotated colorectal tissue images controlled from the gland layout.

3. We demonstrate the efficacy of the annotated pairs generated using the proposed framework for evaluation of the gland segmentation algorithm.

## 2. Materials and Methods

Our aim is to develop the framework to construct the colorectal tissue image having benign/normal grade of cancer and its corresponding tissue component mask highlighting glandular regions, from the input gland layout. The gland layout can be described as a

first quadrant Cartesian plane where users can arrange glands on its 2-d spatial locations. The gland layout is consumed by the framework $f$ that constructs the tissue component mask first, which later assists to generate the complete tissue image of size $N \times N$ pixels. The overview is given in Figure 2. Below we describe the main components of the proposed framework:

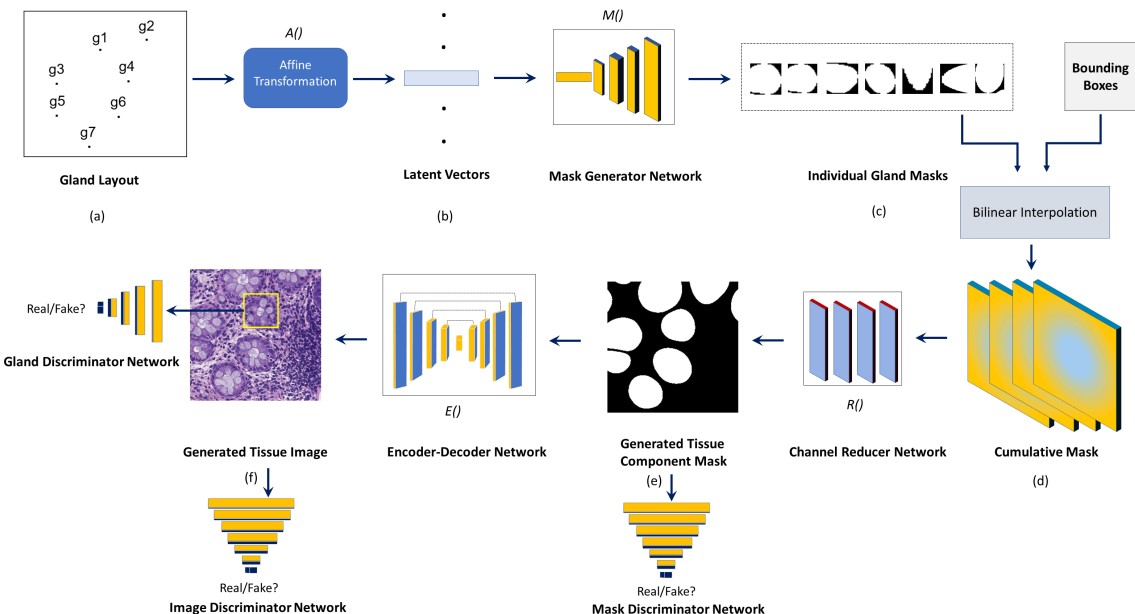

Figure 2: Block diagram of the proposed framework: (a) shows an input gland layout where glands are arranged on 2-d spatial locations. Each gland is characterized by a gland specific vector which undergoes affine transformation to form latent vectors (b). Each latent vector is consumed by the mask generator which outputs binary individual glandular masks (c). The generated masks along with input bounding boxes act as an input to the bilinear interpolation algorithm, which wraps generated masks inside bounding boxes creating the cumulative mask (d). The channel reducer network consumes the cumulative mask and generates the complete tissue component mask (e), which is then passed through the encoder-decoder network generating the final tissue image (f). There are three discriminators employed for generated mask, image and glandular parts inside the image.

The input gland layout can be assumed as a set of $n$ glands, $G_{layout} = \{g_k \equiv (\overrightarrow{l_k}, \overrightarrow{s_k}) \mid k = 1, 2...n\}$, where $\overrightarrow{l_k}$ and $\overrightarrow{s_k}$ denote the location and size vectors respectively of the gland $g_k$. The gland $g_k$ is characterized by a gland specific vector $\overrightarrow{z_k}$ sampled from the Gaussian noise $z \sim \mathcal{N}(0, 1)$. The Gaussian noise is used to ensure the variable appearance of glandular objects in the final image. The gland specific vector $\overrightarrow{z_k}$ is passed through the *affine transformation* $A$, generating $n$ latent embeddings $\{a_k \mid k = 1, 2...n\}$ of dimensionality $D$, i.e., $a_k = A(\overrightarrow{z_k}; \theta_A)$, where $\theta_A$ represents the function's trainable weights.

## 2.1 Generation of Tissue Component Mask

Generated latent embeddings are then consumed by the *mask generator network $M$*, generating the corresponding individual gland binary masks $\{m_k \mid k = 1, 2...n\}$, each of the size $B \times B$ pixels i.e., $m_k = M(a_k; \theta_M)$, where $\theta_M$ denotes the trainable parameters of the network. The mask generator network is comprised of series of blocks having transpose convolution layer followed by the ReLU activation.

Next step is to align generated binary masks on the appropriate locations and construct the tissue component mask of same size as that of the final image, i.e. $N \times N$ pixels. For this purpose, we utilize the input bounding boxes for each object, $\{b_k \mid k = 1, 2...n\}$. These bounding boxes are either obtained from the datasets (procedure given in section 3.1) or realized from the input location and size parameters (section A.3). Each gland specific vector $\vec{z_k}$ is multiplied element-wise with the individual glandular mask $m_k$, and wrapped to the positions of bounding box $b_k$ using the fixed bilinear interpolation function $F$ (Jaderberg et al. (2015)), to give the cumulative mask of dimensionality $D \times B \times B$ , i.e., $C = F(\{\vec{z_k}, m_k, b_k \mid k = 1, 2...n\})$. The cumulative mask has $D$ channels which are then reduced to 1 channel using the *channel reducer network $R$*, to give the tissue component or gland mask, i.e. $T = R(C, \theta_R)$, where $\theta_R$ denotes its trainable parameters.

## 2.2 Tissue Image Generation

After generating the tissue component mask, we feed it to the *encoder-decoder network $E$*. The image-to-image translation encoder-decoder network is used as an image generator, to construct the final tissue image $Z = E(T, \theta_E)$. The encoder consists of a series of (convolution + ReLU) blocks that generate a fixed size encoding of the input image. The decoder constructs the final image using a series of (Transpose convolution + ReLU) blocks. Similar to the generator used in (Isola et al. (2017)), the network $E$ also adopted skip-connections between the layers with the same sized feature maps so that the first downsampling layer is connected with the last upsampling layer, the second downsampling layer is connected with the second last upsampling layer, and so on. These skip-connections give image generator the flexibility to bypass the encoding part to subsequent layers and enable consideration of low level features from earlier encoding blocks in the generator.

## 2.3 Discriminators

We employ 3 discriminator neural networks in an adversarial training setting: *mask discriminator $D_T(T, \theta_{T_D})$* for the generated tissue component mask ($T$), *image discriminator $D_Z(Z, \theta_{Z_D})$* for the generated tissue image $Z$, and the *gland discriminator $D_G(Z_{g_i}, \theta_{G_D})$* for glandular portions $\{Z_{g_i} \mid i = 1, 2..n\}$ inside the tissue image, where $n$ is the number of glands; $\theta_{T_D}$, $\theta_{Z_D}$ and $\theta_{G_D}$ denotes the respective trainable parameters of those discriminators. The first two discriminators employ PatchGAN (Isola et al. (2017)) discriminator which predicts the realism of the different portions from the generated component mask and the tissue image, respectively. The adversarial losses based on these discriminators ensure tissue component masks and tissue images are realistic.

The architecture of the *gland discriminator* is comprised of a series of convolution operations and predicts a single score of realism for the generated glandular portions cropped out from the final tissue image based on input bounding boxes, and resized to a fixed size using bilinear interpolation (Jaderberg et al. (2015)). It ensures the generated glands, one by one, appear real with their micro components like goblet cells and lumen.

We employ an adversarial loss function (Goodfellow et al. (2014)) for all discriminators. A discriminator $D_t(X, \theta_{t_D})$ attempts to maximize the loss by classifying the input image $X$ generated by the generator function $G(X, \theta_G)$ which tries to minimize it, where $\theta_{t_D}$ and $\theta_G$ denotes the set of trainable parameters of the respective networks. $D_t$ is the discriminator type (of $D_T$, $D_Z$ or $D_G$). The adversarial min-max loss function is given by,

$$\min_{\theta_G} \max_{\theta_{t_D}} L_{GAN}^t(\theta_G, \theta_{t_D}) = E_{X \sim p_{data}(X)}[log D_t(X, \theta_{t_D})] + E_{X \sim p_X(X)}[log(1 - D_t(G(X, \theta_G), \theta_{D_t}))]$$
(1)

The training loss of the proposed framework is made up of several terms as it involves multiple networks. Full details about all the loss components and model architectures can be found in Appendices A.1 and A.2, respectively.

## 3. Experiments and Results

In this section, we discuss the process of data collection, model training, and assess the generated images both visually and quantitatively.

### 3.1 Data Acquisition

We use the DigestPath (Li et al. (2019)) colonoscopy tissue segment dataset to assess the performance of our algorithm. The dataset is collected from the DigestPath2019 challenge[1]. It contains 660 very large tissue images with an average size of $5000 \times 5000$ pixels. Each image is associated with pixel-level annotation for glandular regions. This dataset originally contained annotations for malignant lesions (250 images) only. In order to obtain a tissue segmentation mask for benign glands, we used a semi-automatic approach. For this purpose, we first trained a gland segmentation model named Mild-Net (Graham et al. (2019)) on the GlaS dataset (Sirinukunwattana et al. (2017); Sirinukunwattana et al. (2015)), and obtained gland segmentation masks for images with normal grades, in the DigestPath dataset which were manually refined. From these image, we extracted 1733 patches of size $512 \times 512$ that were later resized to $256 \times 256$. Out of these, we kept around 1300 patches for training (train set) and the rest for testing purpose (test set).

The procedure to acquire bounding boxes from the gland masks collected from the digestpath dataset, and also to construct them from the input location $\vec{l}$ and size $\vec{s}$ parameters (gland layout) is given in appendix A.3.

### 3.2 Model Training

To train the whole framework, we set the target tissue image size $N = 256$, input noise dimensionality for the gland specific embeddings, $dim(z) = 6$, latent vector size $D = 32$ and generated per gland size $B = 64$. For the loss function (shown in equation 5), we set $\lambda_1 = \lambda_2 = \lambda_3 = 100.0$ and $\lambda_4 = \lambda_5 = \lambda_6 = 1.0$ after cross-validation tuning.

We train all models using Adam optimizer (Kingma and Ba (2014)) with learning rate $10^{-4}$ and batch size 1 for approx 300K iterations. For each iteration, we first update the generator weights $f$, then update discriminators $D_T$, $D_Z$, and $D_G$. The framework is implemented in Pytorch on an Nvidia Titan X and took almost 2 days for training.

---

1. https://digestpath2019.grand-challenge.org/

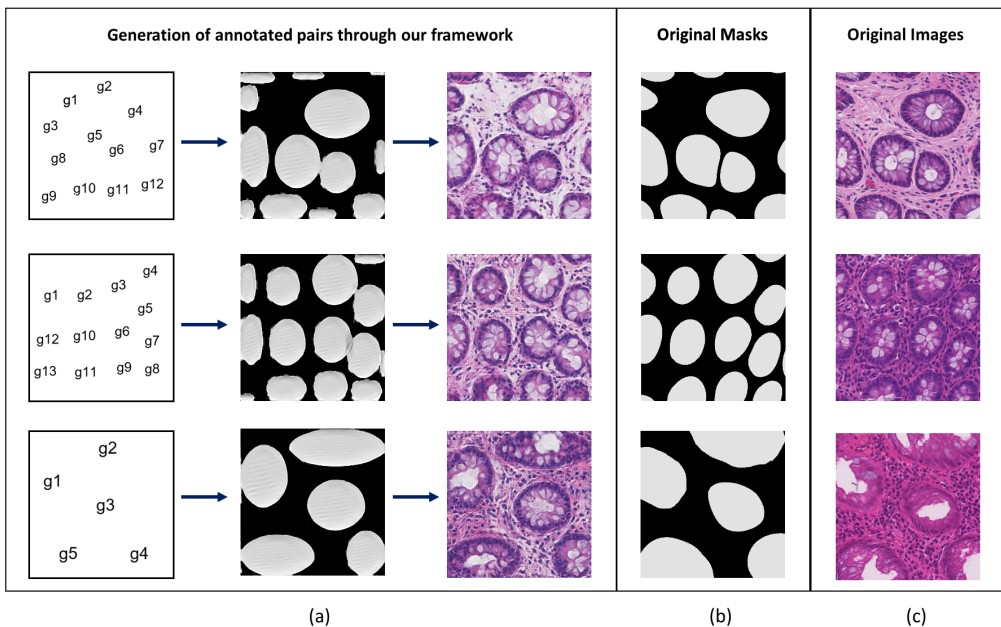

Figure 3: Visual results of generated colorectal tissue images along with their gland segmentation masks from input gland layouts (a). (b) shows original gland segmentation masks while (c) shows the ground truth tissue images.

### 3.3 Visual Results

The visual results of generated tissue images (from the test set), can be seen in Figure 3. We can observe that glandular shape are preserved, tissue components like goblet cells, stromal regions are constructed with fidelity with moderate deformities in the glandular lumen. The generated tissue component masks also appear close to actual masks. The slight deformities and variations in shapes of those masks is a result of using Gaussian noise in the representation embedding of glands, which also make them realistic in nature. We also investigate the change in appearance of glands after altering size $\vec{s}$ and location $\vec{l}$. Figure 4 shows the results of tissue images after changing their locations and sizes. The bounding boxes get modified after altering sizes and locations, which effectively change the size and orientation of glands.

### 3.4 Quantitative Analysis

We evaluate the quality of generated images (from the test set) using the Frechet Inception Distance (FID) (Heusel et al. (2017)), a standard metric used to assess the quality of images by the generative model. It computes the distance between convolution feature maps calculated for real and generated images. For our experiments, to collect convolution features, we use the pretrained InceptionV3 (Szegedy et al. (2016)) network trained on the ImageNet dataset (Deng et al. (2009)). The lower the FID score is the better is the image quality. As the metric depends on the image size, to get a sense of its scale, we also compute the FID score between ground truth images and random noise of the same

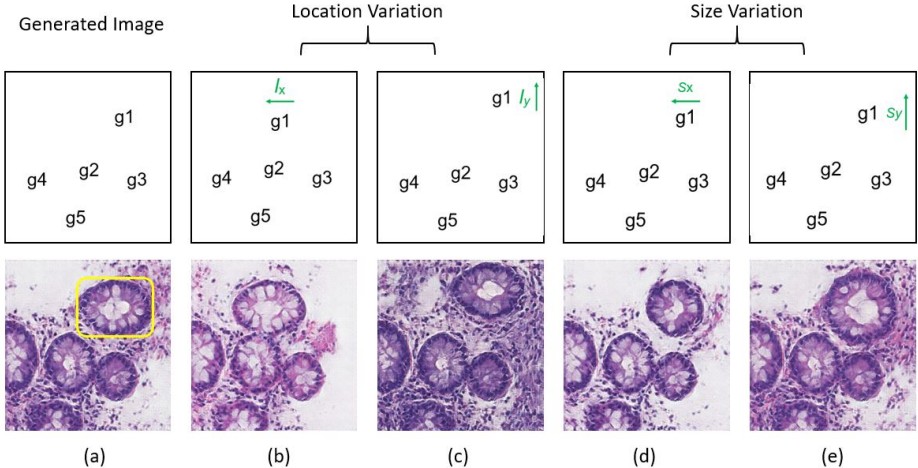

Figure 4: The leftmost image (a) shows the generated sample out from the proposed framework. Images on the right to it shows the change in appearance of the yellow bordered gland, after altering location $\vec{l} = (l_x, l_y)$ and size $\vec{s} = (s_x, s_y)$. (b) and (c) shows the shift of that gland to left side (lowering $l_x$) and upwards (increasing $l_y$), respectively. For the same gland, (d) shows the contraction horizontally and (e) shows expansion vertically after reducing ($s_x$) and increasing ($s_y$), respectively.

size. As a baseline, we adapt the image-to-image translation network Pix2Pix (Isola et al. (2017)) to generate tissue images from existing tissue component masks, and compute the FID for tissue images generated by it. The comparative results are shown in Table 1. Table 1 shows the proposed model achieves better results compared to the random noise and little inferior to that of that of Pix2Pix. The reason can be that as, we are aiming to construct gland segmentation masks as well along with the final tissue images, while Pix2Pix assumes ground truth masks already present and constructs the tissue image from it. Thus, the construction error in generation of masks can influence the performance of our framework, and may slightly lower the quality of generated images. However, looking at the scale of FID values, the difference between both frameworks is not significant.

| Model | FID |
|---|---|
| Random | 485 |
| Pix2Pix | 120 |
| Proposed Framework | 134 |

Table 1: Frechet Inception Distance (FID) score comparison

## 3.5 Assessment through gland segmentation

We also assess the quality of annotated pairs generated by our framework using the U-net based gland segmentation algorithm (Ronneberger et al. (2015)). We train U-net on patches of size $256 \times 256$ from the train set and compute segmentation masks of both real

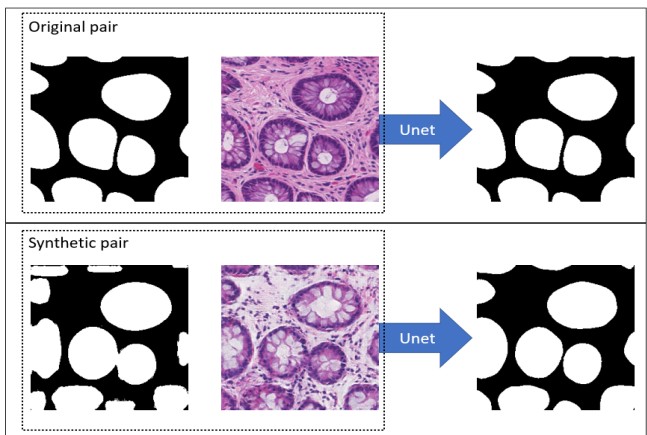

Figure 5: Samples of both real (above) and constructed (below) annotated pairs of tissue images and corresponding gland segmentation masks. The masks shown on the right side are generated from the U-net based segmentation algorithm when applied on original (above) and synthetic (below) images.

and synthetic images from the test set. We use the Dice score (Zou et al. (2004)) between the real component masks and masks computed by U-net on real images, and also between the generated component masks and masks computed by U-net on synthetic images. Sample results are shown in Figure 5.

We obtained an average Dice index of 0.9022 (with standard deviation 0.006) and 0.9001 (with standard deviation 0.012) for both respective cases. This highly similar obtained score validates the applicability of both generated tissue images along with their tissue component masks, for the evaluation of gland segmentation algorithms.

## 4. Conclusion and Future Directions

In this paper, we presented a framework to generate annotated pairs of colorectal tissue images along with their tissue component masks. We performed experiments on DigestPath dataset and demonstrate the framework's ability to generate realistic images preserving morphological features including stroma, goblet cells and glandular lumen. The generated images maintain good FID scores when compared with the state-of-the-art image-to-image translation model. We showed variability in glandular appearance after altering sizes and locations of glands. Additionally, we also demonstrated the applicability of synthetic annotated pairs for the evaluation of gland segmentation algorithms.

The idea can be extended in future to generate annotated tissue image pairs for various tasks in computational histopathology such as nuclei segmentation, cancer grading etc. The generated pairs can potentially replace the real-world training data, pass the legal and security barriers while using them, assist the training and validation of digital pathology algorithms, and reduce the cost and efforts of acquiring data.

# Appendix A.

## A.1 Training Loss Components

Here we give details about all loss components used in our framework. The complete framework with trainable parameters $\{\theta_A, \theta_M, \theta_R, \theta_E, \theta_{T_D}, \theta_{Z_D}, \theta_{G_D}\}$ is trained by minimizing a loss function with the following components:

**Individual Glandular Masks Reconstruction Loss:** This component penalize the difference between ground truth $\{\hat{m}_k \mid k = 1, 2...n\}$ and generated individual binary glandular masks $\{m_k \mid k = 1, 2...n\}$ using the mean square error (MSE) as follows,

$$L_{GlandMaskRec}(\theta_A, \theta_M) = \sum_{k=1}^{n} MSE(\hat{m}_k, m_k) \tag{2}$$

where $\hat{m}_k$ is the ground truth, $m_k$ is the generated individual glandular mask and $n$ is the number of glands in the tissue image. As we saw in section 2, $m_k$ is dependent on the trainable parameters $\theta_A$ and $\theta_M$. Similarly, we can define the other loss components.

**Mask Reconstruction Loss:** This loss is employed to penalize the difference between ground truth $\hat{T}$ and generated tissue component mask $T$ with mean square error (MSE) as below,

$$L_{MaskRec}(\theta_A, \theta_M, \theta_R) = MSE(\hat{T}, T) \tag{3}$$

where $\hat{T}$ is ground truth and $T$ is the generated tissue component mask.

**Image Reconstruction Loss:** This component captures the reconstruction error between ground truth $\hat{Z}$ and generated tissue image $Z$ using the $L1$ difference,

$$L_{ImageRec}(\theta_A, \theta_M, \theta_R, \theta_E) = \|\hat{Z} - Z\|_1 \tag{4}$$

where $\hat{Z}$ is ground truth and $Z$ is the generated tissue image.

**Adversarial Loss Components**: As we discussed in section 2.3, we employ 3 adversarial loss components: $L_{GAN}^T$, $L_{GAN}^Z$ and $L_{GAN}^G$ for tissue component mask, tissue image and the glandular portions cropped out from the tissue image, respectively. Their expressions can be realized by putting $t = T$, $t = Z$ and $t = G$ in (1).

Thus, the overall learning problem can be cast as a the following adversarial optimization problem based on the linear combination of adversarial and reconstruction losses,

$$
\begin{aligned}
\min_{\theta_A, \theta_M, \theta_R, \theta_E} \max_{\theta_{M_D}, \theta_{Z_D}, \theta_{G_D}} & \lambda_1 L_{ImageRec}(\theta_A, \theta_M, \theta_R, \theta_E) + \lambda_2 L_{MaskRec}(\theta_A, \theta_M, \theta_R) \\
& + \lambda_3 L_{GlandMaskRec}(\theta_A, \theta_M) + \lambda_4 L_{GAN}^T(\theta_G = \{\theta_A, \theta_M, \theta_R\}, \theta_{t_D} = \theta_{M_D}) \\
& + \lambda_5 L_{GAN}^Z(\theta_G = \{\theta_A, \theta_M, \theta_R, \theta_E\}, \theta_{t_D} = \theta_{Z_D}) \\
& + \lambda_6 L_{GAN}^G(\theta_G = \{\theta_A, \theta_M, \theta_R, \theta_E\}, \theta_{t_D} = \theta_{G_D})
\end{aligned}
\tag{5}
$$

where $\lambda_1, \lambda_2..\lambda_6$ denote the weights of corresponding loss components.

## A.2 Neural Network Architectures

Here we describe all network architectures for all components in our proposed framework.

## Mask Generator Network

We generate the segmentation mask for each of the glands using the *mask generator network*. The input is the individual gland latent vectors obtained after affine transformation on the original gland embeddings, and output is the $M \times M$ glandular mask with all elements ranged between 0 and 1. The mask regression network composed of series of mask generator blocks, where each block consist of interpolation + convolution + batch normalization + ReLU activation operations. The exact architecture is shown in the table 2, while architecture of the *mask generator block* is shown in Table 3.

| Index | Inputs | Operation | Output Shape |
|-------|--------|-----------|--------------|
| (1) | - | Gland Latent Vector | 32 |
| (2) | (1) | Reshape | 32 x 1 x 1 |
| (3) | (2) | Mask Generator Block | 32 x 2 x 2 |
| (4) | (3) | Mask Generator Block | 32 x 4 x 4 |
| (5) | (4) | Mask Generator Block | 32 x 8 x 8 |
| (6) | (5) | Mask Generator Block | 32 x 16 x 16 |
| (7) | (6) | Mask Generator Block | 32 x 32 x 32 |
| (8) | (7) | Mask Generator Block | 32 x 64 x 64 |
| (9) | (8) | Conv2d (K=1, 32 → 1) | 1 x 64 x 64 |
| (10) | (9) | Sigmoid | 1 x 64 x 64 |

Table 2: Architecture of the mask generator network. The function implements function $M$ from the main text. The notation Conv2d(K , $C_{in} \rightarrow C_{out}$) is a convolution with $K \times K$ kernels, $C_{in}$ input channels and $C_{out}$ output channels; all convolutions with stride 1 with zero padding that ensures input and output have the same spatial size.

| Operation | Output Shape |
|-----------|--------------|
| Interpolation | 32 x 2S x 2S |
| Conv2d (K=3, 32 → 32) | 32 x 2S x 2S |
| Batch Normalization | 32 x 2S x 2S |
| ReLU | 32 x 2S x 2S |

Table 3: Architecture of the mask generator block. The input is the feature map of shape $C \times S \times S$, where C is the number of channels from the feature map of the last layer, and $S \times S$ is the dimension of height and width.

## Channel Reducer Network

The generated cumulative mask (explained in 2.1) has 32 channels, which got reduced to 1 using the *channel reducer network*, forming the tissue component mask. The network comprised of a series of convolution + Relu operations. The exact architecture is shown in Table A.2.

| Index | Inputs | Operation | Output Shape |
|-------|--------|-----------|--------------|
| (1) | - | Generate Cumulative Mask | 32 x 256 x 256 |
| (2) | (1) | Conv2d (K=3, 32 → 16) | 16 x 256 x 256 |
| (3) | (2) | LeakyReLU | 16 x 256 x 256 |
| (4) | (3) | Conv2d (K=3, 16 → 8) | 8 x 256 x 256 |
| (5) | (4) | LeakyReLU | 8 x 256 x 256 |
| (6) | (5) | Conv2d (K=3, 8 → 4) | 4 x 256 x 256 |
| (7) | (6) | LeakyReLU | 4 x 256 x 256 |
| (8) | (7) | Conv2d (K=3, 8 → 4) | 1 x 256 x 256 |
| (9) | (8) | LeakyReLU | 1 x 256 x 256 |

Table 4: Architecture of the channel reducer network. The network implements function $R$ from the main text. LeakyReLU uses a negative slope coefficient of 0.2

## Encoder-Decoder Network

The final tissue image is generated from the generated tissue component mask with the help of *encoder decoder network*. The encoder consist of a series of "Encode" blocks (shown in Table 6) and generates the lower sized encoding of the input mask, while decoder comprised of a series of "Decode" blocks (shown in Table 7) and generates the final tissue image from the encoding. The exact architecture of the encoder-decoder network is shown in Table 5.

## Mask and Image Discriminators

The discriminator we employed has the similar architecture for both tissue component masks ($D_T$) and tissue images ($D_T$), takes the real or fake image of shape $C \times 256 \times 256$ as an input ($C = 1$ for component mask and $C = 3$ for tissue image), and classifies an overlapping grid of size $7 \times 7$ image patches from the input image as real or fake. The exact architecture of the discriminator is shown in Table 8

| Index | Inputs | Operation | Output Shape |
|---|---|---|---|
| (1) | - | Generate Component Mask | 1 x 256 x 256 |
| (2) | (1) | Encode(1,64) | 64 x 128 x 128 |
| (3) | (2) | Encode(64,128) | 128 x 64 x 64 |
| (4) | (3) | Encode(128,256) | 256 x 32 x 32 |
| (5) | (4) | Encode(256,512) | 512 x 16 x 16 |
| (6) | (5) | Encode(512,512) | 512 x 8 x 8 |
| (7) | (6) | Encode(512,512) | 512 x 4 x 4 |
| (8) | (7) | Encode(512,512) | 512 x 2 x 2 |
| (9) | (8) | Encode(512,512) | 512 x 1 x 1 |
| (10) | (9,8) | Decode(512,512) | 1024 x 2 x 2 |
| (11) | (10,7) | Decode(1024,512) | 1024 x 4 x 4 |
| (12) | (11,6) | Decode(1024,512) | 1024 x 8 x 8 |
| (13) | (12,5) | Decode(1024,512) | 1024 x 16 x 16 |
| (14) | (12,4) | Decode(1024,256) | 512 x 32 x 32 |
| (15) | (14,3) | Decode(512,128) | 256 x 64 x 64 |
| (16) | (15,2) | Decode(256,64) | 128 x 128 x 128 |
| (17) | (16) | Upsample | 128 x 256 x 256 |
| (18) | (17) | Conv2d (K=4, 128 $\to$ 3) | 3 x 256 x 256 |
| (19) | (18) | Tanh | 3 x 256 x 256 |

Table 5: Architecture of the encoder-decoder network. The network implements the function $E$ from the main text.

| Operation | Output Shape |
|---|---|
| Conv2d (K=4, $C_{in} \to C_{out}$) | $C_{out}$ x S x S |
| Instance Normalization (if normalize=True) | $C_{out}$ x S x S |
| LeakyReLU | $C_{out}$ x S x S |
| Dropout (if dropout=True) | $C_{out}$ x S x S |

Table 6: Architecture of the "Encode" block. LeakyReLU uses a negative slope coefficient of 0.2

| Operation | Output Shape |
|---|---|
| ConvTranspose2d(K=4, $C_{in} \to C_{out}$) | $C_{out}$ x S x S |
| Instance Normalization | $C_{out}$ x S x S |
| ReLU | $C_{out}$ x S x S |
| Dropout (if dropout=True) | $C_{out}$ x S x S |

Table 7: Architecture of the "Decode" block

**Gland Discriminator**

Our gland discriminator $D_G$ consumes image pixels corresponding to glandular areas from the real or fake tissue images, and classifies them as real or fake. The glandular areas are

| Index | Inputs | Operation | Output Shape |
|:---:|:---:|:---:|:---:|
| (1) | - | Generate the Image | C x 256 x 256 |
| (2) | (1) | Conv2d (K=4, C → 16, S=2) | 16 x 128 x 128 |
| (3) | (2) | LeakyReLU | 16 x 128 x 128 |
| (4) | (3) | Conv2d (K=4, 16 → 32, S=2) | 32 x 64 x 64 |
| (5) | (4) | LeakyReLU | 32 x 64 x 64 |
| (6) | (5) | Instance Normalization | 32 x 64 x 64 |
| (7) | (6) | Conv2d (K=4, 32 → 64, S=2) | 64 x 32 x 32 |
| (8) | (7) | LeakyReLU | 64 x 32 x 32 |
| (9) | (8) | Instance Normalization | 64 x 32 x 32 |
| (10) | (9) | Conv2d (K=4, 64 → 128, S=2) | 128 x 16 x 16 |
| (11) | (10) | LeakyReLU | 128 x 16 x 16 |
| (12) | (11) | Instance Normalization | 128 x 16 x 16 |
| (13) | (12) | Conv2d (K=4, 128 → 256, S=2) | 256 x 8 x 8 |
| (14) | (13) | LeakyReLU | 256 x 8 x 8 |
| (15) | (14) | Instance Normalization | 256 x 8 x 8 |
| (16) | (15) | Conv2d (K=4, 256 → 1 , S=1) | 1 x 7 x 7 |

Table 8: Architecture of the Discriminator network. C=1 when input is the tissue component mask and C=3 for the tissue image. All but the last Conv2d operation has stride 2. LeakyReLU uses a negative slope coefficient of 0.2

| Index | Inputs | Operation | Output Shape |
|:---:|:---:|:---:|:---:|
| (1) | - | Crop glandular portions from the generated image | 3 x 64 x 64 |
| (2) | (1) | Conv2d (K=5, 3 → 16, S=2) | 16 x 30 x 30 |
| (3) | (2) | Batch Normalization | 16 x 30 x 30 |
| (4) | (3) | LeakyReLU | 16 x 30 x 30 |
| (5) | (4) | Conv2d (K=5, 16 → 32, S=2) | 32 x 13 x 13 |
| (6) | (5) | Batch Normalization | 32 x 13 x 13 |
| (7) | (6) | LeakyReLU | 32 x 13 x 13 |
| (8) | (7) | Conv2d (K=5, 32 → 64, S=2) | 64 x 5 x 5 |
| (9) | (8) | Global Average Pooling | 64 |
| (10) | (9) | Affine Transformation | 1024 |
| (11) | (10) | Affine Transformation | 1 |

Table 9: Architecture of the gland discriminator, $D_{gland}$. LeakyReLU uses a negative slope coefficient of 0.2

cropped out using their bounding box coordinates, and resized to $64 \times 64$ pixels using the bilinear interpolation method. The exact architecture of the gland discriminator is shown in Table 9

## A.3 Acquisition of Bounding Boxes

After we collect the tissue images and their annotated gland masks from the digestpath dataset as described in section 3.1, we used the OpenCV (Open Source Computer Vision Library) python library (Bradski (2000)) to extract the location of glands i.e., centroids of white blob objects from the black and white tissue component mask (as shown in mask in figure 1). Later we also collected the bounding boxes of those identified glandular objects using the built-in function boundingRect() function of the same library.

During inference, apart from using the ground truth bounding boxes obtained by the procedure described above, we also construct bounding box for the gland $g_k$ using the input size $\vec{s_k}$ and location $\vec{l_k}$ attributes, taken from the gland layout. Given the input size $\vec{l_k} = (s_{kx}, s_{ky})$, where $s_x$ and $s_y$ are the horizontal and vertical spanning lengths of glands, and the centroid location $\vec{l_k} = (l_{kx}, l_{ky})$, the bounding box coordinates for $g_k$ are computed as,

$$b_k = (l_{kx} - (s_{kx}/2), (l_{ky} - (s_{ky}/2), (l_{kx} + (s_{kx}/2), (l_{ky} + (s_{ky}/2)). \tag{6}$$

Overall, input to the proposed framework is the set of glandular locations, their sizes, their bounding boxes acquired from the dataset or constructed using the above procedure, and output is the pair of the tissue image and its tissue component mask.

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
