# OpenReview forum: "Synthesis of Annotated Colorectal Cancer Tissue Images from Gland Layout"
_MICCAI.org/2021/Workshop/COMPAY — Reject_

### Official Review · Reviewer_9Qoq · 2021-08-09
**A generative method for colonic gland images.**

**Rating:** 5
**Confidence:** 4

**Review:**

The authors describe a method for synthesizing images of glandular structures resembling those on microscopic sections of colon tissues.
Their method uses previously published data from the DigestPath2019 challenge, and is evaluated using a U-Net based gland segmentation algorithm. The paper is methodologically sound.
However, I feel there are two significant weaknesses of the paper with respect to what is claimed.

-The authors claim to synthesize "Annotated Colorectal Cancer Tissue Images". However, all they show is their method can produce glandular-looking structures.
Without being a board-certified pathologist, this reviewer feels that the morphological variety of colorectal cancer is not covered in the generated images shown.
While the generated structures shown superficially "look glandular", the paper does not address at all the grade of dysplasia present on the images, which would be crucial for the generated images to be of practical use.
Additionally, glandular structures are by far not the only morphological structure significant in the assessment of colorectal cancer. Hence, the claim to synthesize "Annotated Colorectal Cancer Tissue Images" seems overly bold.
-A second significant shortcoming of the paper seems the definition of "realistic looking" used thoughout. While the discriminator may produce a "score of realism", this does not show that the images generated are really histologically plausible.
For example, do cells shown in the images possess the correct number of nuclei? Are cell boundaries and tissue context realistic?
While such shortcomings may still allow gland segmentation, they could be immediately obvious as artifacts to a trained pathologist.
Hence, I feel that the claim of "realistic" images can only be maintained when a trained expert cannot distinguish generated images from real samples.

Both points imply significant limitations to the method shown and its practical use, and should be addressed or the claims of the paper modified accordingly.

---

### Official Review · Reviewer_aqTT · 2021-08-19
**The authors of this paper propose a method to generate synthetic colorectal tissue images with glands.**

**Rating:** 5
**Confidence:** 4

**Review:**

The authors of this paper propose a method to generate synthetic colorectal tissue images with glands. I like the main idea of generating synthetic images interactively to train and evaluate algorithms. But the paper could be strengthened by describing more details. Please see my comments below.
1. The authors introduce that this method is user-interactive, but I am curious where the interaction is needed. Based on my understanding of this paper, the user could interact by altering location and size but the proposed framework can still generate realistic colorectal tissue images. Then would a set of gland layout with location/size vectors be manually defined by the user and tuned until the user is satisfied on generated tissue images? A clear description of the interactive component is required. Are there any performance improvement between automatically generated gland layout and interactively generated gland layout?
2. It was difficult for me to understand the method especially from gland specific vector to Channel Reducer Network to generate tissue component mask, which I believe is one of the main contributions of this paper. What exactly is a gland specific vector and why is it needed for affine transformation? I was not able to understand the sentence in the second paragraph of Section 2.1, “Each gland specific vector …” I believe the location vector is two-dimensional specifying x and y coordinates, but what is the size vector composed of? I think one actual example of the location vector, the size vector, the gland specific vector, the affine-transformed latent vector, and cumulative masks on Figure 2 would greatly help the readers to understand these components and why they are needed.
3. More thorough analysis of this method would be desired. For example, what happens if tissue component mask is generated from gland layout without affine transformation/mask generation network/cumulative mask/channel reducer network?
4. In the result section, I think it’s okay to have comparable FID values between Pix2Pix and the proposed framework, but I was hoping to see comparison for Dice score evaluation – Pix2Pix approach is missing here. In addition, the abstract states that synthetic images can be useful in training, but the result of a model trained by synthetic images generated by the proposed method is missing. In my personal opinion, training a model with synthetic images is clinically more important than evaluating a model on synthetic images.

---

### Decision · Program_Chairs · 2021-08-25

Reject